# Physics3D: Learning Physical Properties of 3D Gaussians via Video Diffusion

## Abstract

In recent years, there has been rapid development in 3D generation models, opening up new possibilities for applications such as simulating the dynamic movements of 3D objects and customizing their behaviors. However, current 3D generative models tend to focus only on surface features such as color and shape, neglecting the inherent physical properties that govern the behavior of objects in the real world. To accurately simulate physics-aligned dynamics, it is essential to predict the physical properties of materials and incorporate them into the behavior prediction process. Nonetheless, predicting the diverse materials of real-world objects is still challenging due to the complex nature of their physical attributes. In this paper, we propose **Physics3D**, a novel method for learning various physical properties of 3D objects through a video diffusion model. Our approach involves designing a highly generalizable physical simulation system based on a viscoelastic material model, which enables us to simulate a wide range of materials with high-fidelity capabilities. Moreover, we distill the physical priors from a video diffusion model that contains more understanding of realistic object materials. Extensive experiments demonstrate the effectiveness of our method with both elastic and plastic materials. Physics3D shows great potential for bridging the gap between the physical world and virtual neural space, providing a better integration and application of realistic physical principles in virtual environments. Project page: https://physics3d-3dgs.github.io

## 1 Introduction

In recent years, 3D computer vision has witnessed significant advancements, with researchers focusing on reconstructing or generating 3D assets Mildenhall et al. (2021); Kerbl et al. (2023); Tang et al. (2023); Poole et al. (2022); Hong et al. (2023); Li et al. (2024), and even delving into the realm of 4D dynamics Ren et al. (2023); Ling et al. (2023). However, a common feature in these works is the emphasis on color space, which can be difficult in modeling realistic interactive dynamics without any physical priors, especially for applications in areas such as virtual/augmented reality and animation. Physics simulation is one of the most crucial methods to achieve a deeper understanding of the real world and enhance the effectiveness of interactive dynamics. Although conventional methods Stewart (2000); Felippa (2004); Kilian & Ochsendorf (2005) describe behavior using continuous physical dynamic equations based on body-fixed mesh, they are usually difficult and time-consuming to generate complex 3D objects and suffer from highly nonlinear issues, large deformations or fracture-prone physical phenomenaJiang et al. (2016).

Powered by recent advances in implicit and explicit 3D representation techniques (*e.g.,* NeRF Mildenhall et al. (2021) and 3D Gaussian Splatting Kerbl et al. (2023)), some researchers Xie et al. (2023); Li et al. (2022) have attempted to bridge the gap between rendering and simulation using the differentiable Material Point Method (MPM) Hu et al. (2018b), which enables efficient physical simulation driven by 3D particles (*i.e.,* 3D Gaussian kernels). PhysGaussian Xie et al. (2023) extends the capabilities of 3D Gaussian kernels by incorporating physics-based attributes such as velocity, strain, elastic energy, and stress. This unified representation of material substance facilitates both simulation and rendering tasks. However, manual pre-design of physical parameters in PhysGaussian Xie et al. (2023) remains a laborious and imprecise process, where objects are categorized into six material types: jelly, metal, sand, foam, snow, and plasticine. Each category has distinct physical models and parameters, making it inconvenient to manually classify objects and initialize parameters based on

Figure 1: Physics3D is a unified simulation-rendering pipeline based on 3D Gaussians, which learn physics dynamics from video diffusion model. **The top row** shows the dynamic effects in the 2D RGB video space, which represents the starting of the SDS loss in the optimization process. **The bottom row** shows the dynamic effects in the 3D Gaussian space, representing the final optimization goal of SDS. Each 3D Gaussian, when rendered from a specific viewpoint, corresponds to a frame in the 2D video. The different objects in the two rows are intended for visual contrast and clarity.

expert knowledge. To avoid manually setting parameters, PhysDreamer Zhang et al. (2024) leverages object dynamics learned from video generation models Blattmann et al. (2023b); Wang et al. (2023) to estimate a physical material parameter (*i.e.,* Young's modulus). However, in practical applications, real-world objects often exhibit a complex composite nature, making it challenging for a simulation approach that relies solely on a single physical parameter to fully capture their dynamic behavior. This limits PhysDreamer to be primarily tailored for the simulation of hyper-elastic materials. Specifically, it encounters significant challenges when dealing with materials such as plastics, metals, and non-Newtonian fluids due to its heavy reliance on optimizing Young's modulus alone. The inherent complexities in these materials surpass the capabilities of PhysDreamer, highlighting the need for a more comprehensive and robust approach that considers a broader range of physical properties for accurate and effective simulation.

In this paper, we propose **Physics3D**, a generalizable physical simulation system to learn various physical properties of 3D objects. Given a 3D Gaussian representation, we first expand the dimension of the physical parameters to capture both elasticity and viscosity. Then we design a viscoelastic Material Point Method (MPM) to simulate 3D dynamics. Through the simulation process, we decompose the deformation gradient into two separate components and calculate them independently to contribute to the overall force. Finally, leveraging the capabilities of the differentiable MPM, we iteratively optimize both 3D Gaussian parameters and physical parameters via the Score Distillation Sampling (SDS) Poole et al. (2022) strategy to distill physical priors from the video diffusion model. Iterating the MPM process and SDS optimization, Physics3D achieves high-fidelity and realistic performance in a wide range of materials. Extensive experiments demonstrate the efficacy and superiority of our proposed Physics3D over existing methods. In summary, our key contributions are as follows.

- We propose a novel generalizable physical simulation system called Physics3D, which is capable of learning physical properties of diverse materials. We model physical properties with both elastoplastic and viscoelastic parts and design a parallel simulation framework.

- We design a physics-driven distillation strategy to iteratively optimize both filling 3D Gaussians and physical parameters, realising to generate realistic, physics-driven and controllable 3D dynamics while maintaining the model's generalization capability with limited 3D data.

- Experiments show Physics3D is effective in creating high-fidelity and realistic 3D dynamics, ready for various interactions across users and objects in the future.

## 2 RELATED WORK

**Dynamic 3D representations.** Rapid advancements in static 3D representations have sparked interest in incorporating temporal dynamics into the 3D modeling of dynamic objects and scenes. Various explicit or hybrid representation techniques have demonstrated impressive outcomes, including planar decomposition for 4D space-time grids Cao & Johnson (2023); Shao et al. (2023); Fridovich-Keil et al. (2023), the utilization of NeRF representation Li et al. (2022); Pumarola et al. (2021); Gao et al.

(2021), and alternative structural approaches Turki et al. (2023); Abou-Chakra et al. (2024); Fang et al. (2022). Recently, 3D Gaussian Splatting Kerbl et al. (2023) has revolutionized the representation via its outstanding rendering efficiency and high-quality results. Efforts have been made to extend static 3D Gaussians into dynamic versions, yielding promising results. Dynamic 3D Gaussians Luiten et al. (2023) refine per-frame Gaussian Splatting through dynamic regularizations and shared properties such as size, color, and opacity. Similarly, the concept of 4D Gaussian Splatting Wu et al. (2023); Yang et al. (2023) employs a deformation network to anticipate time-dependent positional, scaling, and rotational deformations. In addition, DreamGaussian4D Ren et al. (2023) learns motion from image-conditioned generated videos Blattmann et al. (2023a), enabling more controllable and diverse 3D motion representations.

**Viscoelastic materials.** In the realm of computer graphics and animation, there has been significant interest in accurately simulating the behavior of nonrigid objects and their interactions with physical environments. Conventional elastic models Terzopoulos et al. (1987); Zong et al. (2023) predicated on Hooke's law Rychlewski (1984) are the cornerstone for simulating the deformation of objects. These models are effective in representing materials that exhibit perfectly elastic behavior, returning to their original shape after the applied force is removed. However, real-world materials often exhibit more complex behaviors that cannot be captured by simple elastic models. The introduction of viscoelastic materials in computer graphics Terzopoulos & Fleischer (1988) has expanded the range of simulated material behaviors. Viscoelastic materials combine the characteristics of both viscous fluids and elastic solids, leading to time-dependent deformations under constant stress, a phenomenon known as creep Christensen (2003). Compared with elastic models, viscoelastic models offer a more versatile framework for animating nonrigid objects. They can simulate the slow restoration of a material's shape after the cessation of force, as well as the permanent deformation that occurs due to prolonged stress.

**Video generation models.** With the emergence of models like Sora Brooks et al. (2024), the field of video generation Villegas et al. (2022); Wu et al. (2022); Bar-Tal et al. (2024); Blattmann et al. (2023c) has drawn significant attention. These powerful video models Kondratyuk et al. (2023); Singer et al. (2022); Ho et al. (2022); Hong et al. (2022) are typically trained on extensive datasets of high-quality video content. Sora Brooks et al. (2024) is capable of producing minute-long videos with realistic motions and consistent viewpoints. Furthermore, some large-scale video models, such as Sora Brooks et al. (2024), can even support physically plausible effects. Inspired by these capabilities, we aim to distill the physical principles observed in videos and apply them to our static 3D objects, thereby achieving more realistic and physically accurate results. In our framework, we choose Stable Video Diffusion Blattmann et al. (2023b) to optimize our physical properties.

## 3 PROBLEM FORMULATION

Given a static representation through 3D Gaussians, our goal is to estimate the physical attributes of each Gaussian particle and generate physics-plausible motions by organizing the interaction of force and velocity among these particles. For pure-elastic models, these physical properties include mass ($m$), Young's modulus ($E$), and Poisson's ratio ($\nu$). Young's modulus and Poisson's ratio control the dynamics of elastic objects. For example, with a fixed amount of external force applied, the system's higher Young's modulus will have smaller deformation.

However, only modeling the property of elastic objects is inadequate for recovering diverse physics with heterogeneous materials in real-world applications, which significantly limits the recent work like PhysDreamer Zhang et al. (2024). For example, they usually suffer from complex mixed materials, especially in scenarios of rapid deformation where viscosity emerges as a significant factor in dynamics. Therefore, our key insight is to build a more comprehensive physics model that includes additional parameters, notably viscosity, to enrich the descriptive capacity for real-world objects, especially in inelastic scenarios.

To model viscoelastic stresses with physical fidelity, we explore continuum mechanics, where Lamé constants (also referred to as Lamé coefficients or parameters), denoted by $\lambda$ and $\mu$, emerge as pertinent material-related quantities within the strain-stress relationship, while viscosity coefficient $\nu_v$ and $\nu_d$ governs the viscous dynamics. Consequently, our framework pivots towards the estimation of the viscosity coefficient ($\nu_v$ and $\nu_d$) and the two Lamé parameters ($\lambda$, and $\mu$). As for other physical attributes, we align with the conventional methods Zhang et al. (2024); Xie et al. (2023), where the

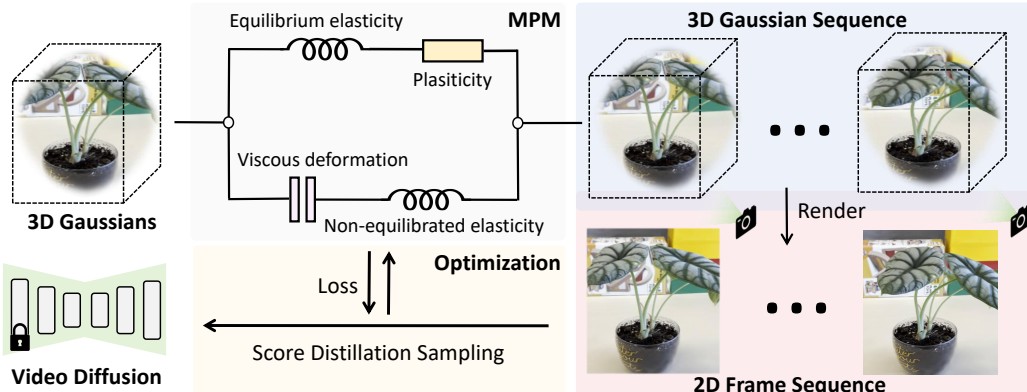

Figure 2: **Pipeline of Physics3D**. Given an object represented as 3D Gaussians, We first simulate it using the Material Point Method (MPM). The simulation comprises two distinct components: an elastoplastic part and a viscoelastic part. These components operate independently to calculate the individual stresses within the object and are combined to determine the overall stress. After the simulation, a series of Gaussians with varying orientations are generated, reflecting the dynamic evolution of the scene. Then, we render these Gaussians from a fixed viewpoint to produce a sequence of video frames. Finally, we utilize a pretrained video diffusion model with Score Distillation Sampling (SDS) strategy to iteratively optimize physical parameters.

particle mass ($m_p$) is pre-calculated as the product of a constant density ($\rho$) and the particle volume ($V_p$), and Poisson's ratio ($\nu$) is constant across the object. For detailed explanation and clarified summary of notations, please refer to Appendix C.1.

# 4 METHOD

In this section, we introduce our method, *i.e.*, Physics3D, for learning the dynamics of multi-material 3D systems with physical alignment. Our goal is to estimate the various physical properties of 3D objects. Building upon this, we first review the theory of three foundational techniques (Sec. 4.1) that form the backbone of our algorithm. Then we introduce our physical modeling framework and describe our particle-based simulation process (Sec. 4.2). Finally, we present the physical-based distillation strategy (Sec. 4.3) to iteratively optimize both filling 3D Gaussians and physical parameters with the video diffusion model. An overview of our framework is depicted in Figure 2.

## 4.1 PRELIMINARY

**Continuum Mechanics** describes motions by a deformation map $\mathbf{x} = \phi(\mathbf{X}, t)$ from the material space $\Omega^0$ (with coordinate $\mathbf{X}$) to the world space $\Omega^n$ (with coordinate $\mathbf{x}$). The deformation gradient $\mathbf{F} = \frac{\partial \phi}{\partial \mathbf{X}}(\mathbf{X}, t)$ measures local rotation and strain Bonet & Wood (1997). We consider viscoelastic materials where we have two components Govindjee & Reese (1997), the elastoplastic component $\mathbf{F_E F_P}$ and the viscoelastic component $\mathbf{F_N F_V}$. They are in parallel combination shown in Figure 3, and formulated as:

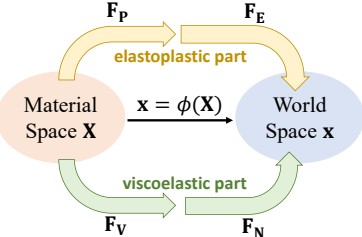

Figure 3: Elastoplastic and viscoelastic decomposition.

$$\mathbf{F} = \mathbf{F_E F_P} = \mathbf{F_N F_V}. \tag{1}$$

Intuitively, we model our material with two different compounds in parallel connection as they deform in the same way thus having the same total strain. However, only the elastic components $\mathbf{F_E}$ and $\mathbf{F_N}$ contributes to the internal stress $\boldsymbol{\sigma}_E$ and $\boldsymbol{\sigma}_N$. We can now evolve the system with dynamical equations. Denoting the velocity field with $\mathbf{v}(\mathbf{x}, t)$ and density field with $\rho(\mathbf{x}, t)$, the conservation of momentum and conservation of mass Germain (1998) is given by:

$$\rho \frac{D\mathbf{v}}{Dt} = \nabla \cdot \boldsymbol{\sigma} + \mathbf{f}, \quad \frac{D\rho}{Dt} + \rho \nabla \cdot \mathbf{v} = 0, \tag{2}$$

where $\mathbf{f}$ denotes an external force, $\boldsymbol{\sigma} = \boldsymbol{\sigma}_E + \boldsymbol{\sigma}_N$ is the total internal stress. We will also need to update the strain tensor after updating the material point.

**Material Point Method (MPM)** Sulsky et al. (1995); Jiang et al. (2016); Hu et al. (2018b) discretizes the material into deformable particles and employs particles to monitor the complete history of strain and stress states while relying on a background grid for precisely evaluating derivatives during force computations. This methodology has demonstrated its efficacy in simulating diverse materials Jiang et al. (2015); Klár et al. (2016); Stomakhin et al. (2013) and is proved to be capable of simulating some viscoelastic and viscoplastic materials Ram et al. (2015); Yue et al. (2015). MPM operates in a particle-to-grid (P2G) and grid-to-particle (G2P) transfer loop. In P2G process, MPM transfers mass and momentum from particles to grids:

$$m_i^n = \sum_p w_{ip}^n m_p, \quad m_i^n \mathbf{v_i^n} = \sum_p w_{ip}^n m_p (\mathbf{v_p^n} + C_p^n(\mathbf{x_i} - \mathbf{x_p^n})), \tag{3}$$

Here $i$ and $p$ represent the fields on the Eulerian grid and the Lagrangian particles respectively. Each particle $p$ carries a set of properties including volume $V_p$, mass $m_p$, position $\mathbf{x_p^n}$, velocity $\mathbf{v_p^n}$, deformation gradient $\mathbf{F_p^n}$ and affine momentum $C_p^n$ at time $t_n$. $w_{ip}^n$ is the B-spline kernel defined on $i$-th grid evaluated at $\mathbf{x_p^n}$. After P2G, the transferred grids can be updated as:

$$\mathbf{v_i^{n+1}} = \mathbf{v_i^n} - \frac{\Delta t}{m_i} \sum_p \tau_p^n \nabla w_{ip}^n V_p^0 + \Delta t \mathbf{g}, \tag{4}$$

here $\mathbf{g}$ represents the acceleration due to gravity. Then G2P transfers velocities back to particles and updates particle states $\tau$ (*i.e.,* Kirchhoff tensor).

$$\tau_p^{n+1} = \boldsymbol{\tau}(\mathbf{F_E^{n+1}}, \mathbf{F_N^{n+1}}), \tag{5}$$

where $\mathbf{F_E^{n+1}}$ and $\mathbf{F_E^{n+1}}$ are two parts of strain tensor. We will provide a detailed introduction in Sec. 4.2. In this work, we integrate the physical properties of viscoelastic materials into the Material Point Method, thereby enhancing the generalization capabilities of MPM. This implementation enables the simulation for a wide range of materials commonly found in the real world, including various inelastic materials. We refer to the Appendix for more details.

**Score Distillation Sampling (SDS)** is first introduced by DreamFusion Poole et al. (2022), which distills the 3D knowledge from large 2D pretrain model Saharia et al. (2022). For a set of particles parameterized by physical properties $\theta$, its rendering $\mathbf{x}$ can be obtained by $\mathbf{x} = g(\theta)$ where $g$ is a differentiable renderer. SDS calculates the gradients of physical parameters $\theta$ by,

$$\nabla_\theta \mathcal{L}_{\text{SDS}}(\phi, \mathbf{x} = \mathbf{g}(\theta)) = \mathbb{E}_{t,\epsilon} \left[ w(t)(\boldsymbol{\epsilon}_\phi(\mathbf{x}_t; y, t) - \boldsymbol{\epsilon}) \frac{\partial \mathbf{x}_t}{\partial \mathbf{x}} \frac{\partial \mathbf{x}}{\partial \theta} \right], \tag{6}$$

where $w(t)$ is a weighting function that depends on the timestep $t$ and $y$ denotes the given condition. $\epsilon_\theta(\mathbf{x}_t, t)$ is autoencoder in diffusion model to estimate the origin distribution of the given noise $\mathbf{x}_t$.

## 4.2 Physical Modeling

Our goal is to generate 3D dynamics for diverse materials, especially inelastic materials and composite materials with more complex motion modeling. An intuitive solution is to capture these viscous effects by directly adding a viscosity part following the current elastic pipeline as PhysDreamer Zhang et al. (2024). However, such a simple series connection of components is prone to coupling of the two dissipative behaviors (elastoplastic and viscoelastic), which increases the complexity of the model simulation and disrupts the original elastic properties of the model.

In our paper, we instead design a novel parallel framework to avoid the coupling issue. We decompose the material space into elastoplastic and an additional viscoelastic part, aiming to unify the simulation of a wide range of materials. It showcases the flexibility of this framework, and by adding more blocks into both the static and dynamical modeling, it can be expected to be widely useful for more complicated scenes. To simulate the physical process with Gaussian particles, we employ a particle-based method MLS-MPM Hu et al. (2018a) as our simulator, and we formalize the simulation process for a single sub-step as follows:

$$\mathbf{x^{n+1}}, \mathbf{v^{n+1}}, \mathbf{F^{n+1}}, C^{n+1} = \mathcal{S}(\mathbf{x^n}, \mathbf{v^n}, \mathbf{F^n}, C^n, \theta, \Delta t), \tag{7}$$

Here, $\theta$ contains the physical properties of all particles: mass $m_i$, Young's modulus $E_i$, Poisson's ratio $\nu_i$, Lamé coefficients $\lambda, \mu$, viscosity coefficient $\nu_{Ni}$ and volume $V_i$. $\Delta t$ denotes the simulation step size. Within the MPM simulation, stress, as depicted in Equation 1, can be divided into two components: one representing elastoplasticity, denoted as $\mathbf{F_E}$, and the other referring to viscoelasticity, denoted as $\mathbf{F_N}$. The two parallel parts are also illustrated in Figure 2, consisting of a parallel combination: (a) an elastic part with a frictional element for plasticity and; (b) a viscous part which is assumed to capture the dissipation with a elastic part. Now, we elaborate on the computation for each component individually.

**Model for elastoplastic part.** We first explain how to compute the internal stress $\boldsymbol{\sigma_E}$ through $\mathbf{F_E}$, which is essential in updating kinematical variables like velocities $v$ through Eq. 2, for the elastic part. We will speak out the rule here but postpone explanations and intuitions in Appendix C.4. As we demonstrate in Sec. 4.1 and Appendix C.3, one can compute the Cauchy stress tensor given the energy function. In this work, we choose the fixed corotated constitutive model for the elastic part, whose energy function is

$$\psi(\mathbf{F_E}) = \psi(\mathbf{\Sigma_E}) = \mu_E \sum_i (\sigma_{E,i} - 1)^2 + \frac{\lambda_E}{2}(\det(\mathbf{F_E}) - 1)^2, \tag{8}$$

where $\mathbf{\Sigma_E}$ is the diagonal singular value matrix $\mathbf{F_E} = \mathbf{U\Sigma_E V^T}$. And $\sigma_{E,i}$ are singular values of $\mathbf{F_E}$. From this energy function, we can compute the Cauchy stress tensor as:

$$\boldsymbol{\sigma_E} = \frac{2\mu}{\det(\mathbf{F_E})}(\mathbf{F_E} - \mathbf{R})\mathbf{F_E^T} + \lambda_E(\det(\mathbf{F_E}) - 1), \tag{9}$$

where $\mathbf{R} = \mathbf{UV^T}$. We now explain how to update the $\mathbf{F_E^n}$ with the velocity field $\mathbf{v^n}$ at $n$th step. For purely elastic case, this can be simply done via:

$$\mathbf{F_E^{n+1}} = (\mathbf{I} + \Delta t \nabla \mathbf{v^n})\mathbf{F^n}, \tag{10}$$

where $\Delta t$ is the length of the time segment in the MPM method. This formula represents the fact that the internal velocity field causes a change in strains.

For the plastic part of the branch, we constrain the singular value of $\mathbf{F_E}$ to sit between $[1 - \theta_c, 1 + \theta_s]$, where $\theta_c$ and $\theta_s$ are learnable parameters quantifying the plasticity. More precisely, in the simulation algorithm, at $n$-th step, $\mathbf{F^n} = \mathbf{F_E^n F_P^n}$. We then compute the internal stress $\boldsymbol{\sigma}(\mathbf{F_E^n})$ and update $\mathbf{F^n}$ to $\mathbf{F^{n+1}}$ with MPM method. Please refer to the Appendix C.4 for more details.

**Model for viscoelastic part.** We now explain the other branch of our model: the viscoelastic part. The total strain now consist of two parts $\mathbf{F} = \mathbf{F_N F_V}$. There are two key features of this brunch. First, only the $\mathbf{F_N}$ contributes to the internal stress. Secondly, $\mathbf{F_V}$ only plays a role in the update rule of $\mathbf{F_N}$. For the first step, we know have a relation between internal stress $\boldsymbol{\sigma_N}$ (or equivalently Kirchoff tensor $\boldsymbol{\tau_N} = \det(\mathbf{F_N})\boldsymbol{\sigma_N}$):

$$\tau_N = 2\mu_N \epsilon_N + \lambda_N \mathrm{tr}(\epsilon_N)\mathbf{I} \tag{11}$$

where we denote $\tau_N$ as a vector of singular value of $\boldsymbol{\tau_N}$, in another word $\boldsymbol{\tau_N} = U\mathrm{diag}(\tau_N)V^T$. And $\epsilon_N$, called log principle Kirchoff tensor, denotes a vector takes the diagonal element of $\log \Sigma_N$, where $\Sigma_N$ as the diagonal singular value matrix $\mathbf{F_N} = U\Sigma_N V^T$. We can again update kinemetical variables afterward. The second step is we need to modify the update rule for $\mathbf{F_N}$ tensor. This boils down to a trial-and-correction procedure where we first update

$$\mathbf{F_{N,tr}^n} = (\mathbf{I} + \Delta t \nabla \mathbf{v^n})\mathbf{F_N^n}, \tag{12}$$

then we modify the trial strain tensor $\mathbf{F_{N,tr}^n}$ by

$$\epsilon_N^{n+1} = A(\epsilon_{N,tr}^n - B\mathrm{tr}(\epsilon_{N,tr}^n) \cdot \mathbf{1}) \tag{13}$$

where $\epsilon_N^{n+1}$ denotes the log principle Kirchoff tensor of $\mathbf{F_N^{n+1}}$. $A$ and $B$ are functions of viscocity parameters $\nu_d$ and $\nu_e$. Please refer to Appendix C.4 for more detailed intuitions and explanations.

### 4.3 PHYSICS-DRIVEN DISTILLATION

Through iterations of the Material Point Method (MPM), we obtain a set of Gaussians evolving over time $t$. The general practice is that in order to optimize physical properties for each 3D Gaussian, we need high-quality 3D Gaussian models and corresponding motion ground truth. This ground truth could come from various sources, such as single-view or multi-view real-captured videos, 4D videos, or ideally, detailed 3D models with comprehensive physical property labels.

However, acquiring such datasets is currently challenging. Inspired by former works like Zhang et al. (2024); Ren et al. (2023) which learn 3D dynamics from 2D videos, we intend to create 3D dynamics that appear as realistic videos when given different initial external forces and rendered from random angles. Such 3D dynamics can be specified as a differentiable video parameterization, where a differentiable generator $g$ (here indicates MPM processor and following Gaussian rasterizer) transforms both physical parameters and Gaussian parameters $\theta$ to create a video $x = g(\theta)$. Therefore, we extend the score distillation sampling (SDS) to iteratively optimize both 3D Gaussian parameters and physical parameters from the video diffusion model. To ensure consistency, the camera remains stationary as we capture and render each Gaussian to produce a reference image $I_t^r$. To supervise and optimize this process, we employ two optional models: image-to-video and text-to-video diffusion models. Then, we use SDS loss to guide our optimization process:

$$\nabla_\theta \mathcal{L}_{\text{SDS}} = \mathbb{E}_{t,p,\epsilon} \left[ w(t)(\epsilon_\phi(I_t^p; t, I_t^r, \Delta p, y) - \epsilon) \frac{\partial I_t^p}{\partial \theta} \right]. \tag{14}$$

Here, $w(t)$ represents a time-dependent weighting function, $\epsilon_\phi(\cdot)$ denotes the predicted noise generated by the 2D diffusion prior $\phi$, $I_t^p$ represents a rendered video frame of diffusion timestep $t$ from a camera pose $p$, $\Delta p$ signifies the relative change in camera pose from the reference camera $r$, and $y$ denotes the given condition (*i.e.,* image or text). Furthermore, we adopt a partial filling strategy like Xie et al. (2023), where internal volumes of select solid objects are optionally filled to augment simulation realisticity. For better rendering quality, we optimize the filled Gaussian along with the SDS process. The optimization objective for filled Gaussians is defined as:

$$\mathcal{L}_{\text{Fill}} = \frac{1}{T} \sum_{t=1}^{T} \lambda ||I_t^p - I_t^r||_2^2. \tag{15}$$

For more details about the learnable internal filling strategy, please refer to Appendix C.5.

## 5 EXPERIMENTS

In this section, we conduct extensive experiments to evaluate our Physics3D and show the comparison results against other methods Xie et al. (2023); Zhang et al. (2024); Ren et al. (2023). We first present our qualitative results and comparisons with baselines (Sec. 5.2). Then we report the quantitative results along with a user study (Sec. 5.3). Finally, we carry out more open settings and ablation studies to further verify the efficacy of our framework design (Sec. 5.4). Please refer to the Appendix for more visualizations and detailed analysis.

### 5.1 EXPERIMENT SETUP

**Datasets.** We evaluate our method for generating diverse dynamics using several sources of input. We choose four real-world static scenes from PhysDreamer Zhang et al. (2024) for fair comparison. Each scene includes an object and a background. The objects include a carnation, an alocasia plant, a telephone cord, and a beanie hat. We also employed NeRF datasets Mildenhall et al. (2021) to evaluate the efficacy of our proposed model. Additionally, we utilize BlenderNeRF Raafat (2023) to synthesize several scenes. For more dataset details, please refer to Append A.2.

**Implementation Details.** In our implementation, we initiate the process by reconstructing 3D Gaussians from multi-view images, establishing a foundational representation of the scene. In complex realistic cases, we undertake a segmentation step to differentiate between the background and foreground elements, focusing solely on the latter for subsequent simulation tasks. Prior to

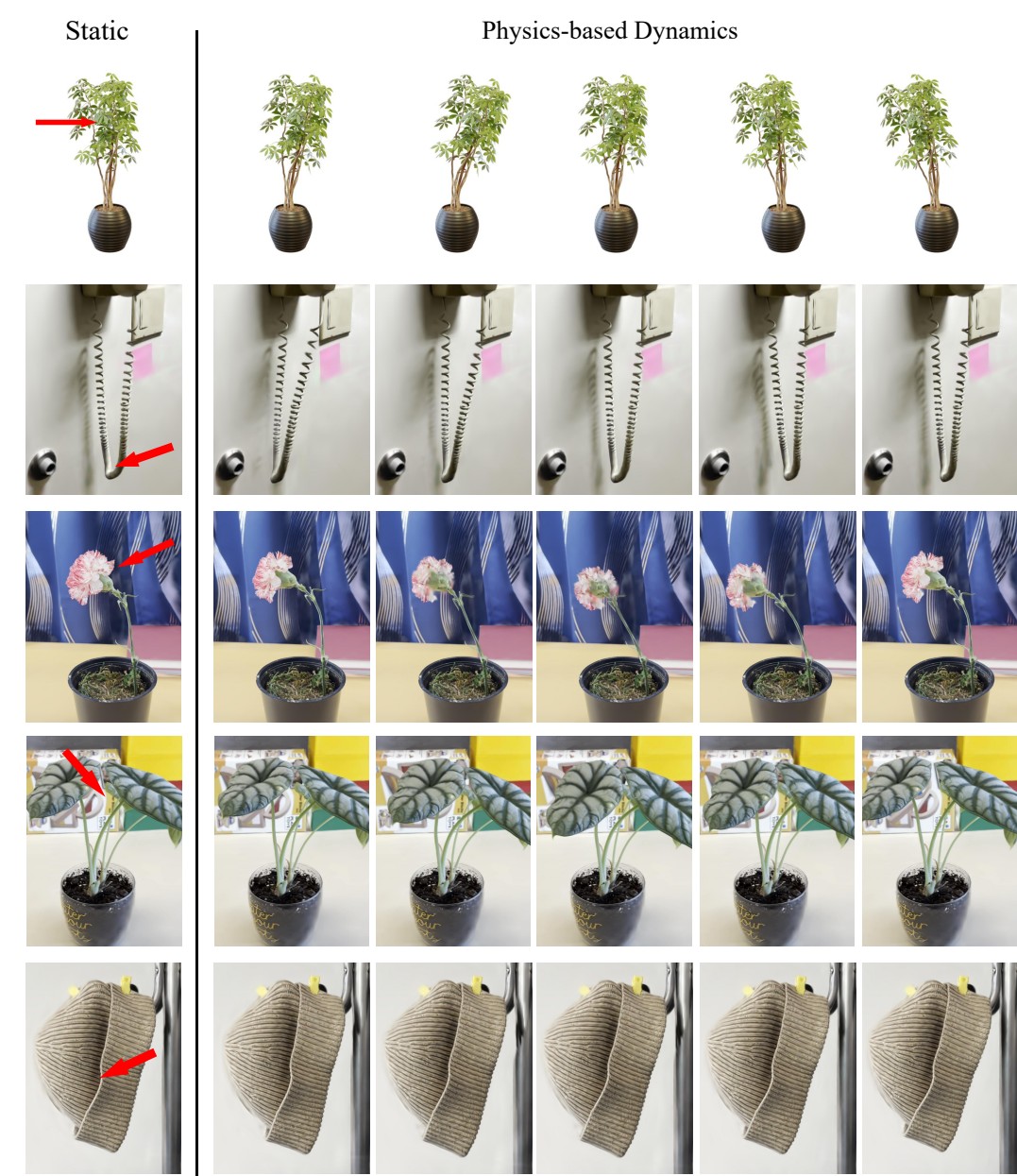

Figure 4: **Visual results of Physics3D** on different subjects with an external force (red arrow). Physics3D is able to generate realistic scene movement while maintaining good motion consistency.

simulation, we execute internal particle filling operations to refine the representation further. Each Gaussian kernel is then associated with a distinct set of physical properties targeted for optimization, facilitating the fine-tuning process. Subsequently, we discretize the foreground region into a grid structure, typically set at dimensions of $50^3$. As for the MPM simulation, we employ 400 sub-steps within each temporal interval spanning successive video frames. This temporal granularity translates to a sub-step duration of $1e-4$ second, ensuring precision and accuracy in the simulation dynamics. Notably, the optimization process for each object requires approximately 5 minutes to complete on a single NVIDIA A6000 (48GB) GPU.

**Baselines and Metrics.** We extensively compare our method with three baselines: Phys-Dreamer Zhang et al. (2024), PhysGaussian Xie et al. (2023) and DreamGaussian4D Ren et al. (2023). Following Zhang et al. (2024), we show our results with notable comparisons with space-time slices. For metrics, we evaluate our approach with the video-quality metrics : PSNR, SSIM, MS-

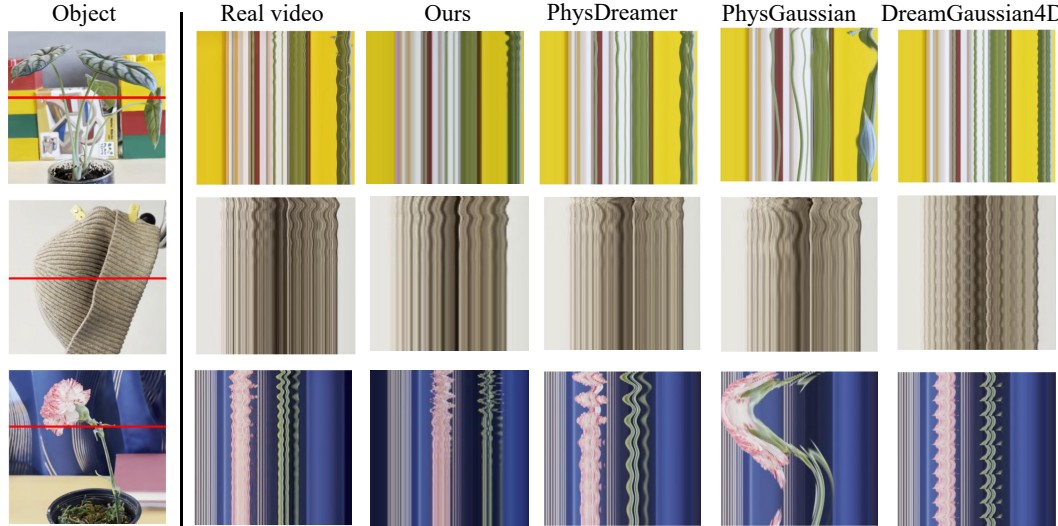

| Object | Real video | Ours | PhysDreamer | PhysGaussian | DreamGaussian4D |

Figure 5: **Comparison results** on different subjects. From the visual results, we observe that PhysDreamer Zhang et al. (2024) only estimates the elastic properties of objects, resulting in the lack of damping. DreamGaussian4D Ren et al. (2023) and PhysGaussian Xie et al. (2023) are respectively limited in unrealistically constant, low-magnitude periodic motion and low-frequency movements. In contrast, our model successfully balances high and low-frequency oscillations with more realistic damping.

SSIM, and VMAF on the rendered videos. Currently, there is a lack of comprehensive 3D dynamic evaluation metrics. PSNR, SSIM and MS-SSIM Free Software Foundation, Inc. (1991) are used to evaluate the quality of generated frames, while video-specific metrics (VMAF Netflix, Inc. (2020)) are employed to assess the dynamic effects of the generated sequences.

## 5.2 QUALITATIVE RESULTS

Figure 4 shows qualitative results of simulated interactive motion. For each case, we visualize one example with its initial scene and deformation sequence. The results demonstrate our model's capability of simulating the movement of complex textured objects, presenting a realistic and physically plausible outcome. Additional experiments are included in the Appendix B.1. Please visit our project page: `https://physics3d-3dgs.github.io` for more dynamic visual results.

Following PhysDreamer Zhang et al. (2024), we compare our results with real captured videos and simulations from other methods Zhang et al. (2024); Xie et al. (2023); Ren et al. (2023) in Figure 5. We utilize space-time slices to present our comparisons. These slices depict time along the vertical axis and spatial slices of the object along the horizontal axis, as indicated by the red lines in the "object" column. Through these visualizations, we aim to elucidate the magnitude and frequencies of the oscillating motions under scrutiny. PhysDreamer Zhang et al. (2024) closely approximates the elastic properties of objects, resulting in periodic oscillations with subtle damping, contrasting the unrealistic aspects. PhysGaussian Xie et al. (2023) showcases unrealistically low-frequency movements due to inaccurate parameter settings. DreamGaussian4D Ren et al. (2023) lacks physical prior and generates unrealistically constant, low-magnitude periodic motion. In contrast, our model successfully balances high and low-frequency oscillations with more realistic damping, aligning more closely with the behavior of objects in the real world.

## 5.3 QUANTITATIVE RESULTS

Table 1 shows the average video-quality metrics over rendered videos from the same fixed perspective. Results clearly demonstrate higher scores for Physics3D, indicating better video

Table 1: **Quantitative comparisons** on rendered videos using different video-quality metrics.

| | PSNR↑ | SSIM↑ | MS-SSIM↑ | VMAF↑ |
|---|---|---|---|---|
| PhysDreamer Zhang et al. (2024) | 13.89 | 0.55 | 0.37 | 0.52 |
| PhysGaussian Xie et al. (2023) | 13.86 | 0.57 | 0.39 | 0.59 |
| Ours | **14.72** | **0.59** | **0.49** | **0.59** |
| - *w/o* viscoelastic part | 14.13 | 0.53 | 0.36 | 0.52 |
| - *w/o* elastoplastic part | 13.53 | 0.55 | 0.41 | 0.50 |

quality and motion consistency of our
results. We also conduct numerous user experiments in Appendix B.3.

## 5.4 ABLATION STUDY

We conduct ablation study in Figure 6 to evaluate the efficacy of our physics modeling. Specifically, we investigate the importance of elastoplastic and viscoelastic components from the model architecture. We observe that removing either of the modules leads to a notable degradation in the realism of the physical simulations. Particularly, the absence of the elastoplastic component results in a lack of elasticity, making objects more susceptible to shape deformation and fluid-like behavior. On the other hand, the absence of the viscoelastic component leads to a deficiency in sustained damping and rebound effects, especially in scenarios with minimal external disturbances where energy dissipation occurs rapidly. These show the significance of both components of our model in capturing the dynamics of physical objects. Please refer to our Appendix B.2 for more ablations.

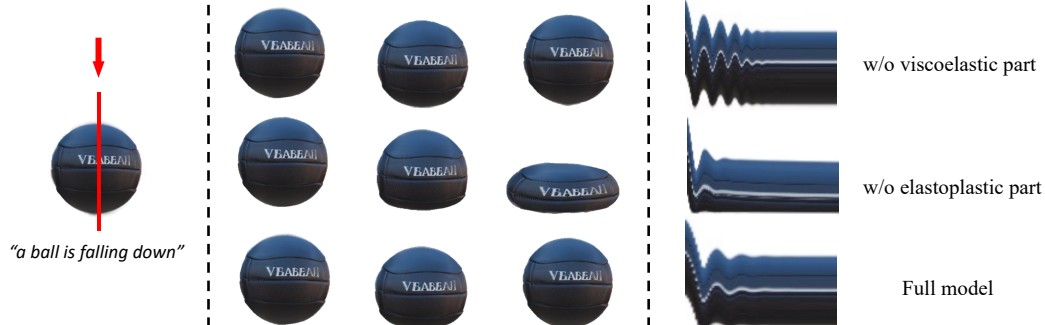

Figure 6: Ablation study on elastoplastic and viscoelastic parts of our model.

## 6 CONCLUSION

In this paper, we present Physics3D, a novel framework to learn various physical properties of 3D objects from video diffusion model. Our method tackles the challenge of estimating diverse material properties by incorporating two key components: elastoplastic and viscoelastic modules. The elastoplastic component facilitates simulations of pure elasticity, while the viscoelastic module introduces damping effects, crucial for capturing the behavior of materials exhibiting both elasticity and viscosity. Furthermore, we leverage a video generation model to distill inherent physical priors, improving our understanding of realistic material characteristics. Extensive experiments show Physics3D is effective in creating high-fidelity and realistic 3D dynamics.

**Limitations and Future Work.** In complex environments with a lot of entangled objects, our method requires manual intervention to assign the scope of movable objects and define the filling ranges for objects, which is not efficient for more real applications. In the future, we aim to utilize the prior of large segmentation models to solve the problem with more comprehensive physics system modeling. We believe that Physics3D takes a significant step to open up a wide range of applications from realistic simulations to interactive virtual experiences and will inspire more works in the future.

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

APPENDIX

# A  ADDITIONAL IMPLEMENTATION DETAILS

## A.1  TRAINING DETAILS.

During the iterative optimization process, we use a specific video diffusion model in our experiment. For the image-to-video model, we utilize Stable Video Diffusion Blattmann et al. (2023b) that learns rich 2D diffusion prior from large real-world data. For a text-to-video diffusion model, we use a diffusion-based model text-to-video-ms-1.7b Wang et al. (2023), which augments the UNet structure with a cross-attention mechanism, which is an effective approach to condition the visual content on texts Huang et al. (2023). Following the text-to-video model, we use the text embedding of the prompt in the spatial attention block as the key and value in the multi-head attention layer. This enables the intermediate UNet features to aggregate text features seamlessly, thereby facilitating an alignment of language and vision embeddings. To ensure a great alignment between language and vision, the text encoder from pre-trained CLIP ViT-H/14 is used to convert the prompt into text embedding. In our experiment, we use text descriptions of object motion as conditional inputs instead of reference images. For example, in Figure 4, the top row uses a text prompt "a ficus swaying in the wind" to generate the video sequence.

In our experiment, we give external force to different objects. For synthetic data, we use a uniform, randomly initialized force for each method. For real-world data in PhysDreamer Zhang et al. (2024), due to the absence of explicit initial external forces and for fair comparisons with PhysDreamer, we choose to estimate an initial force based on the initial velocity field predicted by PhysDreamer. This force is manually tested to perform equally to the motion with the initial velocity field in PhysDreamer. Then we apply the estimated initial force across all baselines in real-world data.

For the representation of the physical parameters to be learned, We do not utilize the tri-plane Chan et al. (2022) approach which is employed in PhysDreamer Zhang et al. (2024). The tri-plane method, while useful, involves a degree of data compression that can compromise the quality of the material representation. Instead, we assign specific physical parameters to each Gaussian kernel directly because of our pursuit of higher fidelity in modeling.

## A.2  DATASET DETAILS

For real-world data, we are leveraging the data in Figure 4 from PhysDreamer Zhang et al. (2024) (carnation, alocasia, hat, telephone cord), which is indeed sourced from real-world scenarios. Due to the limited availability of 3D data and its complexities, following the approach of PhysDreamer allows us to verify the effectiveness of Physics3D on real-world data. Our work in this area continues to need more comprehensive 3D datasets, whether synthetic or real.

For a fair comparison with PhysDreamer Zhang et al. (2024) and others, we use the ground truth videos captured from Physdreamer to compute the reconstruction metrics. Since synthetic 3D data lacks dynamic ground truth, we use it simply for qualitative visual results. We also conduct a user study to further demonstrate the overall quality and the alignment with real-world physics in our simulation, which can be found in Figure 9.

# B  MORE RESULTS

To further demonstrate the effectiveness and impressive visualization results of our Physics3D, we conducted more experiments including additional visual results and user study.

## B.1  ADDITIONAL VISUAL RESULTS

Figure 7 shows additional visual results of interactive video sequences. We use BlenderNeRF Raafat (2023) to synthesize the cases below. For each case, We apply external forces of different directions and magnitudes to static objects and render video frames to show their motion states.

For more complex scenes, we have done more experiments on complex synthetic data such as collisions or fluid dynamics. Please refer to Figure 8, 9 for visual results of complex synthetic data. We are also planning to construct more complex real-world dynamic scenes using the segmentation method to separate objects and their movements, thereby enabling a more detailed analysis of dynamic behaviors. While the scarcity of diverse and high-quality 3D data poses remains challenging, we are making efforts to expand our dataset and refine our methodology. We are still committed to advancing our research by incorporating more complex synthetic and real-world data.

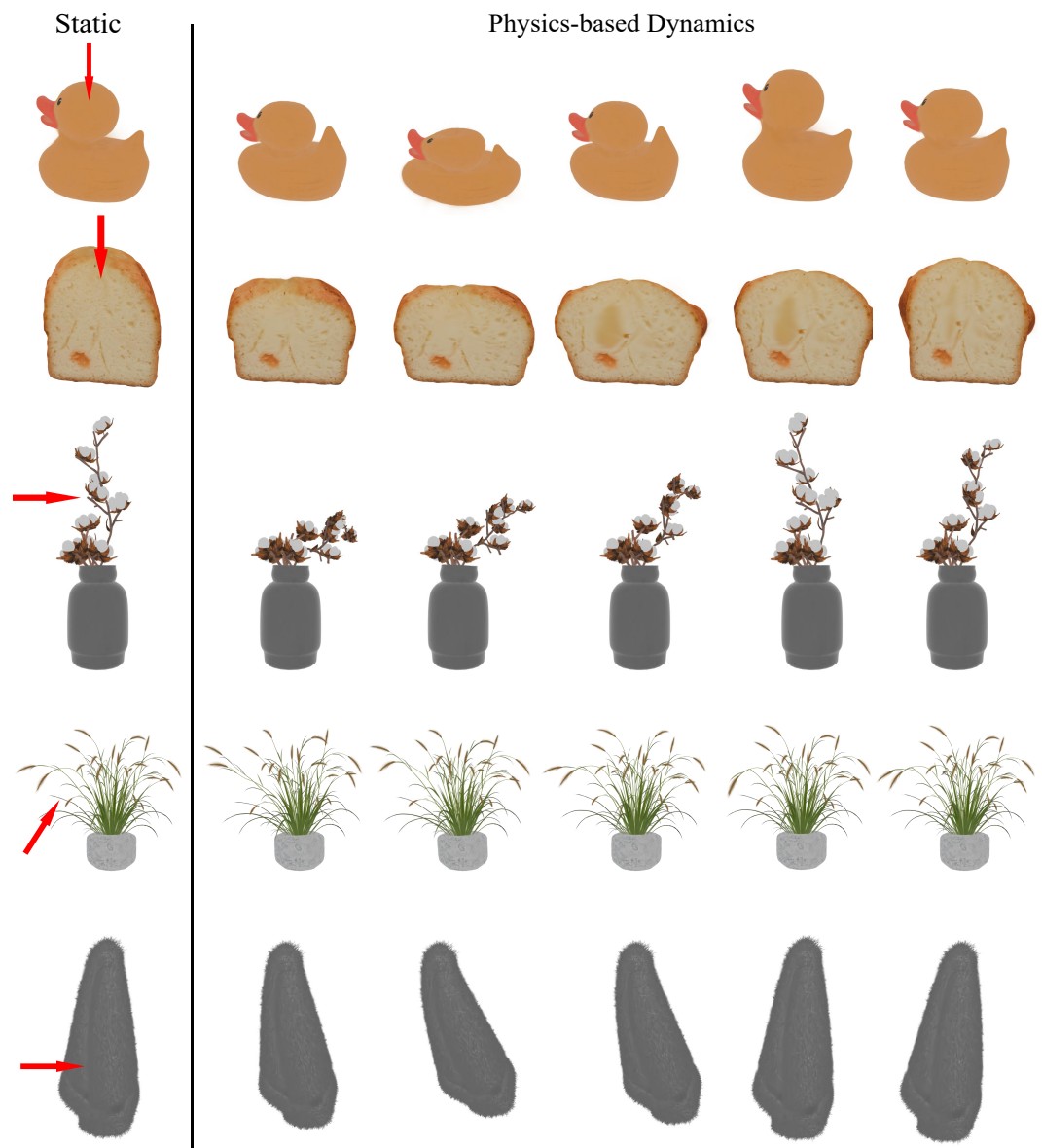

Figure 7: More visual results on different objects. Notice ours can simulate a variety of non-elastic and composite materials, including *rubber*, *fluffy bread*, *fabric* etc.

## B.2 ADDITIONAL ABLATION STUDY

We carry out additional ablation analyses on the Physics3D design in Figure 6 to assess the efficacy of our physics modeling process. Specifically, we investigate the impact of removing the elastoplastic and viscoelastic components from the model architecture. The findings highlight that the removal of either of these modules leads to a notable decrease in the realisticity of the physical simulations.

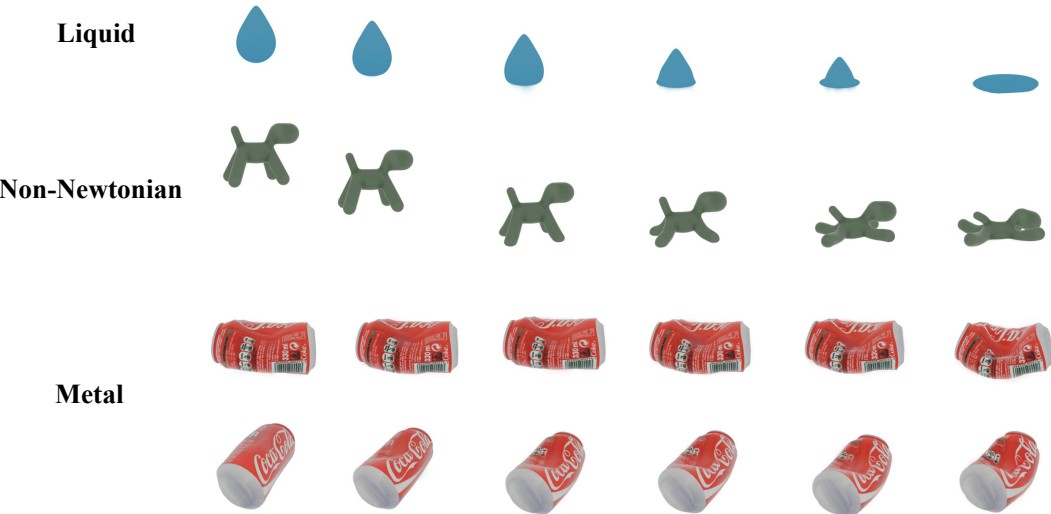

Liquid

Non-Newtonian

Metal

Figure 8: **More visual results of various materials** such as *liquid*, *non-Newtonian fluids* and *metal (a coke can is pressed)*.

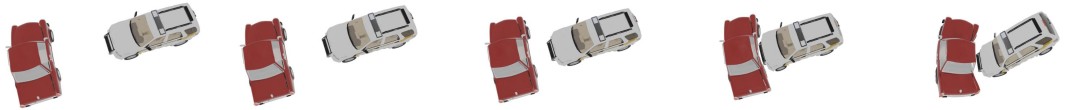

Figure 9: **Variety of dynamic interactions** such as *car collision*.

Particularly, the absence of the elastoplastic component results in a lack of elasticity, thereby making objects more susceptible to shape deformation and fluid-like behavior. On the other hand, the absence of the viscoelastic component leads to a deficiency in sustained damping and rebound effects, especially in scenarios with minimal external disturbances where energy dissipation occurs rapidly. These outcomes underscore the significance of both components of our model in capturing the dynamics of physical objects.

We also conduct an ablation study on the internal filling strategy shown in Figure 10. Notice that without internal filling, the hollow part of the cake is prone to collapse in the simulation process, but with internal filling, the object's deformation under external force is more realistic and smooth. For static and learnable internal filling, we can find that learnable internal filling can significantly reduce noise and artifacts introduced by initialization errors of internal Gaussian particles compared to static internal filling in Figure 11.

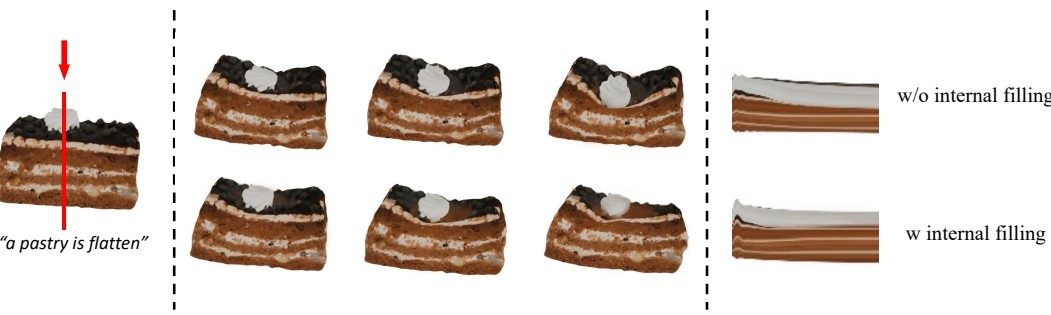

*"a pastry is flatten"*

w/o internal filling

w internal filling

Figure 10: Ablation study on internal filling.

Figure 11: Comparison on static and learning internal filling. **Top row:** static filling strategy encounters challenges in dynamic situations, such as *tearing a bread*, where blur internal regions are exposed to the outside. **Bottom row:** learnable internal filling with iterative optimization strategy of SDS ensures thorough optimization and stability.

### B.3 USER STUDY

For user study, we show each volunteer with five samples of rendered video from a random method (PhysDreamer Zhang et al. (2024), PhysGaussian Xie et al. (2023), DreamGaussian4D Ren et al. (2023), and ours) and real-captured videos. The study engaged 30 volunteers to assess the generated results in 20 rounds. In each round, they were asked to select the video they preferred the most, based on quality, realisticity, alignment with input 3D object, and fluency. We find our method is significantly preferred by users over these aspects.

Table 2: **Quantitative comparison results** of PhysGaussian Xie et al. (2023), PhysDreamer Zhang et al. (2024) and our Physics3D on the action coherence, motion realism, and overall quality score in a user study, rated on a range of 1-10, with higher scores indicating better performance.

| Method | Action Coherence | Motion Realism | Overall Quality |
|---|---|---|---|
| PhysGaussian Xie et al. (2023) | 7.82 | 6.89 | 6.93 |
| PhysDreamer Zhang et al. (2024) | 8.76 | 7.73 | 7.89 |
| Physics3D (Ours) | **8.95** | **8.57** | **9.05** |

## C MORE ANALYSIS

### C.1 PROBLEM FORMULATION AND NOTATION

In modeling viscoelastic stresses with physical fidelity, $\lambda$ and $\mu$ are respectively denoted as Lamé's first and second parameters. Their significance varies in different contexts. For example, in fluid dynamics, $\mu$ is related to the dynamic viscosity of fluids while in elastic environments, Lamé parameters $\lambda$ and $\mu$ intertwine with Young's modulus ($E$) and Poisson's ratio ($\nu$) via a specific relationship. On the other hand, the viscosity coefficient corrects the elastic strain-stress relation by dynamically controlling the relation to account for viscosity. With a closer look at the viscosity coefficient $\nu_v$ and $\nu_d$, it serves as a viscous damper to exert resistance against rapid deformation, thus enhancing the model's fidelity in capturing viscoelasticity. We also summarize the main physical parameters needed in the simulation process in Table 3.

### C.2 MATERIAL POINT METHOD (MPM) ALGORITHM

The Material Point Method (MPM) simulates the behavior of materials by discretizing a continuum body into particles and updating their properties over time. Here's an additional summary of the MPM algorithm:

Table 3: **Material Parameters.**

| Notation | Meaning | Value |
|---|---|---|
| $m$ | Mass of the Gaussian particle. | / |
| $E$ | Young's Modulus: Controls the dynamics of elastic objects; higher $E$ results in smaller deformation under a fixed external force. | Learnable parameter |
| $\nu$ | Poisson's Ratio: Determines the relationship between lateral strain and longitudinal strain in elastic materials. | / |
| $\lambda_E$ | Lamé's First Parameter: Relevant in continuum mechanics for stress-strain relationships in elastoplastic part. | $\lambda_E = \frac{E\nu}{(1+\nu)(1-2\nu)}$ |
| $\mu_E$ | Lamé's Second Parameter (Shear Modulus): Related to elastic materials in elastoplastic part. | $\mu_E = \frac{E}{2(1+\nu)}$ |
| $\nu_d$ | Dynamic Viscosity Coefficient: Serves as a viscous damper exerting resistance against rapid deformation. | Learnable parameter |
| $\nu_v$ | Kinematic Viscosity Coefficient: Governs the dynamic aspect of viscosity in the model for the viscoelastic part. | Learnable parameter |
| $\mu_N$ | Lamé's First Parameter: Relevant in continuum mechanics for stress-strain relationships in viscoelastic part. In fluid dynamics, related to dynamic viscosity. | Learnable parameter |
| $\lambda_N$ | Lamé's Second Parameter (Shear Modulus): Related to elastic materials in viscoelastic part. | Learnable parameter |

**Particle to Grid Transfer.** Mass and momentum are transferred from particles to grid nodes. This step involves distributing the particle properties (mass and velocity) to nearby grid points.

$$m_i^n = \sum_p w_{ip}^n m_p,$$

$$m_i^n \mathbf{v}_i^n = \sum_p w_{ip}^n m_p(\mathbf{v_p^n} + \mathbf{C}_p^n(\mathbf{x_i} - \mathbf{x_p^n})).$$

**Grid Update.** Grid velocities are updated based on external forces and the forces from neighboring particles. This step moves the grid points according to the applied forces.

$$\mathbf{v_i^{n+1}} = \mathbf{v_i^n} - \frac{\Delta t}{m_i} \sum_p \boldsymbol{\tau}_p^n \nabla w_{ip}^n V_p^0 + \Delta t \mathbf{g}.$$

**Grid to Particle Transfer.** Velocities are transferred back to particles, and particle states are updated. This step brings the changes in grid velocities back to the particles.

$$\mathbf{v_p^{n+1}} = \sum_i \mathbf{v_i^{n+1}} w_{ip}^n,$$

$$\mathbf{x_p^{n+1}} = \mathbf{x_p^n} + \Delta t \mathbf{v_p^{n+1}},$$

$$\boldsymbol{C}_p^{n+1} = \frac{12}{\Delta x^2(b+1)} \sum_i w_{ip}^n \mathbf{v_i^{n+1}} \left(\mathbf{x_i^n} - \mathbf{x_p^n}\right)^T,$$

$$\nabla \mathbf{v_p^{n+1}} = \sum_i \boldsymbol{v}_i^{n+1} \nabla w_{ip}^{n\ T},$$

$$\tau_p^{n+1} = \boldsymbol{\tau}(\mathbf{F_E^{n+1}}, \mathbf{F_N^{n+1}}),$$

Here $b$ is the B-spline degree, and $\Delta x$ is the Eulerian grid spacing. We extensively demonstrate how to update $\mathbf{F_E}$, $\mathbf{F_N}$ and $\tau$ in Appendix C.3.

## C.3 DYNAMIC 3D GENERATION.

Dynamic 3D generation aims to synthesize the dynamic behavior of 3D objects or scenes in the process of generating three-dimensional representations. Unlike static 3D generation methods Liu et al. (2023); Hong et al. (2023); Wang et al. (2024); Liu et al. (2024) that solely focus on the spatial morphology of objects, the field of 3D dynamics imposes a higher requirement, necessitating the incorporation of information from all three spatial dimensions as well as the temporal dimension. As advancements in 3D representation techniques continue to emerge, certain parameterizable 3D representations have empowered us to inject dynamic information into 3D models Li et al. (2022); Kratimenos et al. (2023). One popular approach Newcombe et al. (2015); Ren et al. (2023) integrates three essential components:

**Gaussian Splatting** Kerbl et al. (2023) represents 3D information with a set of 3D Gaussian kernels. Each Gaussian can be described with a center $x \in \mathbb{R}^3$, a scaling factor $s \in \mathbb{R}^3$, and a rotation quaternion $q \in \mathbb{R}^4$. Additionally, an opacity value $\alpha \in \mathbb{R}$ and a color feature $c \in \mathbb{R}^3$ are for volumetric rendering. These parameters can be collectively denoted by $\theta$, with $\theta_i = \{x_i, s_i, q_i, \alpha_i, c_i\}$ representing the parameters for the $i$-th Gaussian. The volume rendering color $C$ of each pixel is computed by blending $N$ ordered points overlapping the pixel:

$$C = \sum_{i \in N} c_i \alpha_i \prod_{j=1}^{i-1} (1 - \alpha_j). \tag{16}$$

**Diffusion Model** Ho et al. (2020); Nichol & Dhariwal (2021) is usually pre-trained on large 2D datasets to provide a foundational motion prior for dynamic 3D generation. These models, characterized by their probabilistic nature, are tailored to acquire knowledge of the data distribution $p(x_0)$ through a step-by-step denoising process applied to a normally distributed variable. Throughout the training phase, the data distribution is perturbed towards an isotropic Gaussian distribution over $T$ timesteps, guided by a predefined noising schedule $\alpha_t \in (0, 1)$, where $\overline{\alpha_t} = \sum_{s=1}^{t} \alpha_s$ and $t$ uniformly sampled from $1, ..., T$:

$$\mathbf{z}_t = \sqrt{\bar{\alpha}_t}\mathbf{x_0} + \sqrt{1 - \bar{\alpha}_t}\boldsymbol{\epsilon}, \text{ where } \boldsymbol{\epsilon} \sim \mathcal{N}(\mathbf{0}, \mathbf{I}). \tag{17}$$

The backward denoising process estimates the origin distribution by autoencoder $\epsilon_\theta(\mathbf{z}_t, t)$. The final training loss can be simplified as:

$$\mathcal{L}_{DM} = \mathbb{E}_{\mathbf{x}, \boldsymbol{\epsilon} \sim \mathcal{N}(\mathbf{0}, \mathbf{I})} \left[ \|\boldsymbol{\epsilon} - \boldsymbol{\epsilon}_\phi(\mathbf{z}_t, t)\|_2^2 \right]. \tag{18}$$

**Score Distillation Sampling** (SDS) is a technique used in machine learning, particularly in the context of generative models, to improve sample quality and diversity. It addresses the challenge of generating high-quality samples from complex probability distributions, such as those learned by deep neural networks.

In traditional generative models like Generative Adversarial Networks (GANs) Goodfellow et al. (2020) or Variational Autoencoders (VAEs) Kingma & Welling (2013), generating samples involves directly sampling from the learned latent space. However, this approach often results in low-quality samples with poor diversity, especially in regions of low probability density.

SDS introduces a novel sampling strategy that leverages the notion of score matching. Score matching, introduced by Hyvärinen in 2005 Hyvärinen & Dayan (2005), is a technique for training generative models by matching the score function (gradient of the log-density) of the model distribution to that of the true data distribution.

Given a target distribution $p_{\text{data}}(\mathbf{x})$ and a model distribution $p_\phi(\mathbf{x})$, where $\phi$ represents the parameters of the model, the score function is defined as:

$$\nabla_{\mathbf{x}} \log p_\phi(\mathbf{x}) \tag{19}$$

The score function provides valuable information about the local geometry of the probability distribution, helping to guide the sampling process towards regions of high probability density.

In SDS, the goal is to distill the score function learned by a complex generative model into a simpler, more tractable form. This distilled score function can then be used to guide the sampling process efficiently, resulting in higher-quality samples with improved diversity.

Formally, given a set of generated samples $\{\mathbf{x}_i\}_{i=1}^{N}$ from the model distribution $p_\phi(\mathbf{x})$, the distilled score function $\hat{s}(\mathbf{x})$ is learned to approximate the score function of the model distribution. This is achieved by minimizing the score matching loss:

$$\mathcal{L}_{\text{SM}}(\hat{s}) = \frac{1}{N} \sum_{i=1}^{N} \|\nabla_{\mathbf{x}} \log p_\theta(\mathbf{x}_i) - \hat{s}(\mathbf{x}_i)\|^2 \tag{20}$$

where $\|\cdot\|$ denotes some norm (e.g., $L^2$ norm) and $\hat{s}(\mathbf{x}_i)$ is the estimated score at sample $\mathbf{x}_i$.

As for particular diffusion model, its score function can be related to the predicted noise (shown in Eq. 18) for the smoothed density through Tweedie's formula Robbins (1992):

$$\boldsymbol{\epsilon}_\phi(\mathbf{z}_t, t) = -\sigma_t s_\phi(\mathbf{z}_t; t). \tag{21}$$

Training the diffusion model with a (weighted) evidence lower bound (ELBO) simplifies to a weighted denoising score matching objective for parameters $\phi$ (Ho et al., 2020; Kingma et al., 2021):

$$\mathcal{L}_{\text{Diff}}(\phi, \mathbf{x}) = \mathbb{E}_{t\sim\mathcal{U}(0,1),\epsilon\sim\mathcal{N}(\mathbf{0},\mathbf{I})} \left[ w(t) \|\boldsymbol{\epsilon}_\phi(\alpha_t \mathbf{x} + \sigma_t \epsilon; t) - \epsilon\|_2^2 \right], \tag{22}$$

where $w(t)$ is a weighting function that depends on the timestep $t$. To understand the difficulties of this approach, consider the gradient of $\mathcal{L}_{\text{Diff}}$:

$$\nabla_\theta \mathcal{L}_{\text{Diff}}(\phi, \mathbf{x} = g(\theta)) = \mathbb{E}_{t,\epsilon} \left[ w(t) \underbrace{(\hat{\epsilon}_\phi(\mathbf{z}_t; y, t) - \epsilon)}_{\text{Noise Residual}} \underbrace{\frac{\partial \hat{\epsilon}_\phi(\mathbf{z}_t; y, t)}{\partial \mathbf{z}_t}}_{\text{U-Net Jacobian}} \underbrace{\frac{\partial \mathbf{x}}{\partial \theta}}_{\text{Generator Jacobian}} \right] \tag{23}$$

where following DreamFusion Poole et al. (2022), we absorb the constant $\alpha_t \mathbf{I} = \partial \mathbf{z}_t / \partial x$ into $w(t)$. In practice, the U-Net Jacobian term is expensive to compute (requires backpropagating through the diffusion model U-Net), and poorly conditioned for small noise levels as it is trained to approximate the scaled Hessian of the marginal density. In Poole et al. (2022), It is found that omitting the U-Net Jacobian term leads to an effective gradient:

$$\nabla_\theta \mathcal{L}_{\text{SDS}}(\phi, \mathbf{x} = g(\theta)) \triangleq \mathbb{E}_{t,\epsilon} \left[ w(t) (\hat{\epsilon}_\phi(\mathbf{z}_t; y, t) - \epsilon) \frac{\partial \mathbf{x}}{\partial \theta} \right], \tag{24}$$

here, we get the gradient of a weighted probability density distillation loss in Eq. 14.

## C.4 ADDITIONAL ANALYSIS IN CONTINUUM MECHANICS

**Basic intuitions in continuum mechanics** involves explaining the strain tensor $\mathbf{F}$ which describes the internal deformation of the material, and $\boldsymbol{\sigma}$ which describe the internal stress tensor. In an actual material, there might be multiple compounds, each of them might have their own strain and stress tensor and they might interact with each other. For purpose of this subsection, we explain the intuition when there is only one such compounds. However we will generalize it into multiple compounds case later in the paper.

One could describe an equilibrium material as a point cloud $\mathbf{X}$. In actually simulation, one take a discretization of the system. For our convenience, we will just state the intuition in continuum, and generalization to discrete points should be straightforward. When the material is away from its equilibrium position, the position of point is given by $\mathbf{x} \neq \mathbf{X}$. However, we notice that if, for example, we move $\mathbf{x}$ and the adjacent point $\tilde{\mathbf{x}}$ in the same way, then there is actually no internal deformation of the material. In another word, internal deformation describe how relative position between two points $\mathbf{x}$ and its adjacent $\tilde{\mathbf{x}}$ change. To make this quantity well-defined, we normalize it with the original difference between $\mathbf{X}$ and $\tilde{\mathbf{X}}$

$$\tilde{\mathbf{F}}_{ij} = \frac{\tilde{\mathbf{x}}_\mathbf{i} - \mathbf{x}_i}{\tilde{\mathbf{X}}_\mathbf{j} - \mathbf{X}_j} \tag{25}$$

Taking continuum limit, the equation becomes differential and reproduce

$$\mathbf{F} = \frac{\partial \phi}{\partial \mathbf{X}}(\mathbf{X}, t) \tag{26}$$

in the main text.

To get the dynamics, we will further need the stress tensor $\boldsymbol{\sigma}$. However, the general relation between $\boldsymbol{\sigma}$ and $\mathbf{F}$ is usually complicated. In fact only when internal stress is a conservative force, we can related it to $\mathbf{F}$ through an energy function $\psi(\mathbf{F})$. An analog for this case is the Hooke's force in a spring, where we can introduce a potential energy for the force. There are two main types of internal stress people use in the literature. The first Piola-Kirchoff stress

$$\mathbf{P} = \frac{\partial \psi}{\partial \mathbf{F}} \tag{27}$$

and a related Cauchy stress

$$\boldsymbol{\sigma} = \frac{1}{\det(\mathbf{F})} \frac{\partial \psi}{\partial \mathbf{F}} \mathbf{F}^T \tag{28}$$

We will use Cauchy stress for most part of out paper. However, we could sometimes use the other one interchangeably. Physically, if we want the force acting on a unit surface with normal vector $\hat{n}$, the force will be $\boldsymbol{\sigma} \cdot \hat{n}$. Another version people use is the Kirchoff stress $\boldsymbol{\tau} = \det(\mathbf{F})\boldsymbol{\sigma}$. The only difference between them are whether it is measured with the deformed volume($\boldsymbol{\sigma}$) or the undeformed volume($\boldsymbol{\tau}$). Sometimes using one or the other could be easier for technical reason, as we can see later in the viscoelastic model.

Beyond this simple case, the relation will be quite complicated and non-universal. In this paper, we discuss a specific variant of them, which we explain below. One can see both the relation between $\boldsymbol{\sigma}$ and $\mathbf{F}$, and the update rule for $\mathbf{F}$ get modified.

With these background, we can explain the intuition behind Eq.2. The second equation

$$\frac{D\rho}{Dt} + \rho \nabla \cdot \mathbf{v} = 0, \tag{29}$$

is nothing but mass conservation, $\rho \nabla \cdot \mathbf{v}$ is the local density times the local divergence, which physically correspond to the mass flows out of a local unit volumn in unit time.

The first equation is Newton's law

$$\rho \frac{D\mathbf{v}}{Dt} = \nabla \cdot \boldsymbol{\sigma} + \mathbf{f}, \tag{30}$$

the $\mathbf{f}$ term is the external force. The term $\nabla \cdot \boldsymbol{\sigma}$ is the divergence for the stress tensor, thus giving the total force on a unit volume material.

**Model for elastoplastic part.** We explain how to compute the internal stress $\boldsymbol{\sigma}_E$ through $\mathbf{F_E}$, and how to update $\mathbf{F_E}$ for the elastic part in this appendix. Again for current paper we consider the elastoplastic branch with only elastic part. As we have reviewed, one can compute the Cauchy stress tensor given the energy function. In this work, we choose the Fixed corotated constitutive model for the elastic part, whose energy function is

$$\psi(\mathbf{F_E}) = \psi(\boldsymbol{\Sigma_E}) = \mu \sum_i (\sigma_{E,i} - 1)^2 + \frac{\lambda}{2}(\det(\mathbf{F_E}) - 1)^2 \tag{31}$$

in which $\boldsymbol{\Sigma_E}$ is the diagonal singular value matrix $\mathbf{F_E} = \mathbf{U}\boldsymbol{\Sigma_E}\mathbf{V^T}$. And $\sigma_{E,i}$s are singular values of $\mathbf{F_E}$. From this energy function, one can compute the Cauchy stress tensor

$$\boldsymbol{\sigma_E} = \frac{2\mu}{\det(\mathbf{F_E})}(\mathbf{F_E} - \mathbf{R})\mathbf{F_E^T} + \lambda(\det(\mathbf{F_E}) - 1) \tag{32}$$

in which $\mathbf{R} = \mathbf{U}\mathbf{V^T}$.

We now explain how to update the $\mathbf{F_E^n}$ with the velocity field $\mathbf{v^n}$ at $n$th step. For purely elastic case, this can be simply done via

$$\mathbf{F_E^{n+1}} = (\mathbf{I} + \Delta t \nabla \mathbf{v^n})\mathbf{F^n} \tag{33}$$

in which $\Delta t$ is the length of the time segment in the MPM method. This formula represents the fact that internal velocity field cause change in strains. However, as one will say for the viscoelastic part, this rule will change.

For the plastic part of the branch, we have $\mathbf{F} = \mathbf{F_E}\mathbf{F_P}$, where we constraint the singular value of $\mathbf{F_E}$ to sit between $[1 - \theta_c, 1 + \theta_s]$, where $\theta_c$ and $\theta_s$ are learnable parameters quantifying the plasticity.

More precisely, in the simulation algorithm, at $n$-th step, $\mathbf{F^n} = \mathbf{F^n_E}\mathbf{F^n_P}$. We then compute the internal stress $\boldsymbol{\sigma}(\mathbf{F^n_E})$ and update $\mathbf{F^n}$ to $\mathbf{F^{n+1}}$ with MPM method. At step $n+1$, we first have a trial

$$\tilde{\mathbf{F}}^{\mathbf{n+1}}_{\mathbf{E}} = \mathbf{F^{n+1}}(\mathbf{F^n_P})^{-1}. \tag{34}$$

This $\tilde{\mathbf{F}}^{\mathbf{n+1}}_{\mathbf{E}}$ might have singular value lies out of $[1 - \theta_c, 1 + \theta_s]$, so we truncate the singular value again to get $\mathbf{F^{n+1}_E}$, and get

$$\mathbf{F^{n+1}_P} = \mathbf{F^{n+1}}(\mathbf{F^{n+1}_E})^{-1}. \tag{35}$$

This procedure updates all strain tensor $\mathbf{F_E}, \mathbf{F_P}$ from step $n$ to step $n+1$. Noticing a key difference here between $\mathbf{F_E}$ and $\mathbf{F_P}$ is only $\mathbf{F_E}$ contributes to the stress tensor. This is a manifestation of plasticity, where a large enough deformation will not contribute forces that move the object back to its original position.

**Model for viscoelastic part.** We now explain the other branch of our model: the viscoelastic part. Now we should imagine a elastic strain series connect with a viscous dissipator. In another word, the total strain $\mathbf{F} = \mathbf{F_N}\mathbf{F_V}$. We now explain two key features for this brunch. First, only the $\mathbf{F_N}$ contributes to the internal stress. Secondly, $\mathbf{F_V}$ only plays a role in the update rule of $\mathbf{F_N}$. Very roughly, one can view the dissipator $\mathbf{F_V}$ as a reservoir for strain and each time during the update, people will be able to relocate part of the strain into the $\mathbf{F_V}$, such that only part of the strain contributes to the internal stress.

For the $\mathbf{F_N}$ part, we again take it to be a elastic system, so the relation between strain and stress is again given by the energy function. Here we choose the energy function following Fang et al. (2019)

$$\psi_N(\Sigma_N) = \mu_N \text{tr}((\log \Sigma_N)^2) + \frac{1}{2}\lambda_N(\text{tr}(\log \Sigma_N))^2 \tag{36}$$

In this formula, we introduce $\Sigma_N$ as the diagonal singular value matrix $\mathbf{F_N} = U\Sigma_N V^T$. As one can see, this potential is only a function of the singular values, so the Kirchoff tensor $\boldsymbol{\tau_N}$ will be diagonal in the same basis as the strain $\mathbf{F_N}$. So we will just write down the relation between $\mathbf{F_N}$ and $\boldsymbol{\tau_N}$ just in the singular value basis

$$\tau_N = \frac{\partial \psi_N(\Sigma_N)}{\partial \epsilon_N} = 2\mu_N \epsilon_N + \lambda_N \text{tr}(\epsilon_N)\mathbf{I} \tag{37}$$

where we denote $\tau_N$ as a vector of singular value of $\boldsymbol{\tau_N}$, in another word $\boldsymbol{\tau_N} = U\text{diag}(\tau_N)V^T$. This is sometimes called principle Kirchoff tensor in literature. And $\epsilon_N$, called log principle Kirchoff tensor, denotes a vector takes the diagonal element of $\log \Sigma_N$. So this relation is essentially a vector equation. We can recover the Kirchoff tensor $\boldsymbol{\tau_N}$ thus the Cauchy stress tensor

$$\boldsymbol{\sigma_N} = \frac{1}{\det(\mathbf{F_N})}\boldsymbol{\tau_N}. \tag{38}$$

Now we can discuss the update rule for the strain tensor $\mathbf{F_N}$. As we explained before, the rough intuition was part of the strain tensor $\mathbf{F_N}$ can dissipate into the dissipator $\mathbf{F_V}$. More quantitatively, we can follow the dissipator model in Fang et al. (2019), which results into a trial-and-correction procedure. The idea is we first imagine $\mathbf{F_N}$ evolve elastically

$$\mathbf{F^n_{N,tr}} = (\mathbf{I} + \Delta t \nabla \mathbf{v^n})\mathbf{F^n_N} \tag{39}$$

Next step, we modify this trial strain tensor by introducing a dissipation into the dissipator

$$\epsilon_N^{n+1} = \epsilon_{N,\text{tr}}^n - \Delta t \frac{\partial \psi_V}{\partial \tau_N} \tag{40}$$

Here again, we assume the dissipation happen to only the singular value of the strain tensor, so $\epsilon_{N,\text{tr}}^n$ is the log principle Kirchoff tensor of $\mathbf{F^n_{N,tr}}$ and $\psi_V$ is the dissipation potential. We take a model where

$$\psi_V(\tau_N) = \frac{1}{2\nu_d}|\text{dev}(\tau_N)|^2 + \frac{1}{9\nu_v}(\tau_N \cdot \mathbf{1})^2 \tag{41}$$

where $\nu_d$ and $\nu_v$ are parameters controlling the dissipation of the deviatoric and dilational parts. Physically, parameter $\nu_d$ controls dissipation in deviatoric deformation (deformation that does not change volume, but only the shape), and $\nu_e$ controls dissipation in dilational deformation (deformation

that only changes the volume but not the shape). These two parameters will be the same when materials have a certain kind of homogenous property. We will not use them directly instead, we can put this equation back into Eq.40, and get a version of this formula purely in terms of $\epsilon_N^{n+1}$ and $\epsilon_{N,\text{tr}}^n$

$$\epsilon_N^{n+1} = A(\epsilon_{N,\text{tr}}^n - B\text{tr}(\epsilon_{N,\text{tr}}^n) \cdot \mathbf{1}) \tag{42}$$

In principle, we can write the parameters $A, B$ in terms of $\nu_d, \nu_v$, however since in our algorithm, all these parameters will be learnt from videos, we will directly learn parameter $A$ and $B$ in the update rule without bothering $\nu_d, \nu_v$. But it worth remembering our model comes from a dissipation potential.

### C.5 LEARNABLE INTERNAL FILLING

In Xie et al. (2023), it employs a static filling strategy where the filled particles inherit the parameters $\sigma_p$ and $C_p$ from their nearest Gaussian kernels. These parameters correspond to the Gaussian properties of the filled particles. However, this static filling strategy encounters challenges in dynamic situations, such as tearing or compression shown in Figure 10, where large internal regions are exposed to the outside. The issue arises because the internal particles are not well-initialized and remain fixed during optimization.

To address these challenges, we propose an iterative optimization strategy that refines both physical parameters and internal Gaussian parameters. This approach ensures that the distribution of these two components does not interfere with each other, thereby preventing mode collapse. In other words, one set of parameters can be optimized independently, avoiding "self-sacrifice" for the optimization of the other set. Please refer to the visual comparison of the original and learnable strategy in Figure 11. Specifically, for each epoch in the iterative optimization, we first fix all physical parameters and focus on optimizing the internal filling Gaussian parameters (e.g., Gaussian center $x \in \mathbb{R}^3$, an opacity value $\alpha \in \mathbb{R}$, a color feature $c \in \mathbb{R}^3$ and so on) by Eq. 15. Subsequently, we fix the Gaussian parameters and optimize the physical properties (e.g., Young's modulus $E$, Poisson's ratio $\nu$ , and so on) of each Gaussian by Eq. 14. This iterative process is designed to ensure thorough optimization and stability. Moreover, to enhance the generalizability, we choose a random view for each epoch. This ensures that our model is robust and performs well across different perspectives.

## D ETHICAL STATEMENT

We confirm that all data used in this paper for research and publication have been obtained and used in a manner compliant with ethical standards. The individuals engaged in all experiments have given consent for their use, or the data are sourced from publicly available datasets and were used in accordance with the terms of use and permissions. Furthermore, the publication and use of these data and models do not pose any societal or ethical harm. We have taken necessary precautions to ensure that the research presented in this paper respects individual rights, including the right to privacy and the fundamental principles of ethical research conduct.

