# OpenReview forum: "Physics3D: Learning Physical Properties of 3D Gaussians via Video Diffusion"
_ICLR.cc/2025/Conference — ICLR 2025 Conference Withdrawn Submission_

### Official Review · Reviewer_qSQd · 2024-10-26

**Soundness:** 3
**Presentation:** 3
**Contribution:** 3
**Rating:** 8
**Confidence:** 4

**Summary:**

Physics3D is a framework to simulate physical properties of 3D objects. It learns elastic and viscous behaviors. Authors use viscoelastic MPM with video diffusion model to obtain realistic physical properties. It performs iterative optimization to improve physical behavior and rendering quality. It creates high-fidelity animations of 3D Gaussian representations. Experimental results compared Physics3D with PhysDreamer and PhysGaussian, especially for simulating elasticity and viscosity.

**Strengths:**

- The first to combine MPM viscoelastoplasticity with Gaussians and video diffusion.

- The resulting pipeline is useful and could inspire future research.

- The paper is well written and easy to understand for someone with a continuum mechanics / MPM background.

**Weaknesses:**

- The technical contribution is limited. Many technical pieces are directly taken from previous work. Such as the entire MPM inelasticity simulator.

**Questions:**

- Maybe put text guidance somewhere in Figure 2?

- Section 4.1 should have more references to Fang et al 2019, the main literature that did this viscoelastoplasticity splitting for MPM. The same goes for C.4. These pieces are basically exactly the same with Fang et al 2019. A contribution of this paper is the introduction of it into the ML community.

- Section 4.2 should clarify that the clampping elastoplasticity model comes from Stomakhin et al 2013. It should also clarify that this model is non-physical and invented for snow animation. A more physically grounded elastoplasticity model would be models like von Mises and Drucker Prager, with proper references of mpm literature for those models.

- I’m not exactly getting Figure 11. The bread has not been torn apart. Why do the internal particles matter?

- What’s the text input for fig 4 video diffusion model?

- line 426: “1e-4” -> “10^{-4}”

- The comparisions with PhysGaussian needs to specify the parameters used and how they are chosen. It is unfair to pick a bad parameter that makes the motion bad, and call the method or the model  “unrealistic”. You can say that the manual tuning is time-consuming, and diffusion models allow you to automatically get good parameters. Please modify similar statments about physgaussian throughout the paper.

- I don’t quite understand the experienmnt in Figure 6. The ball is supposed to be elastic wihtout permanent deformations. Why do you need to turn on plasticity for this one?  (A metal or a cake would make more sense for this study.)

- What’s the differentiable MPM used here? Taichi? Warp?

---

### Official Review · Reviewer_nU98 · 2024-10-31

**Soundness:** 3
**Presentation:** 3
**Contribution:** 2
**Rating:** 5
**Confidence:** 2

**Summary:**

This paper introduces Physics3D, a method for learning the physical properties of 3D objects through a video diffusion model. Physics3D employs a parallel simulation framework that integrates both elastoplastic and viscoelastic components, enabling the learning of diverse material properties. Additionally, the authors propose a physics-driven distillation strategy to iteratively optimize both the 3D Gaussian representations and physical parameters. Experimental results demonstrate that Physics3D outperforms existing methods.

**Strengths:**

The paper is well-written and organized. By introducing the viscoelastic component, Physics3D achieves more realistic 3D dynamics and effectively learns the physical properties of various materials. Both quantitative and qualitative results demonstrate that Physics3D outperforms other baselines in generating realistic 3D dynamics. The inclusion of the viscosity component enhances its applicability to real-world materials.

**Weaknesses:**

1. Compared to existing methods, Physics3D only introduces an independent viscoelastic component, which somewhat limits its novelty.
2. The authors report only average quantitative results across all scenes, providing per-scene results would strengthen the credibility of the findings.
3. While the authors argue that "directly adding a viscosity component to the elastic pipeline can lead to coupling between elastoplastic and viscoelastic behaviors, increasing simulation complexity and altering the original elastic properties," this claim is not supported by experimental evidence. Conducting corresponding ablation studies would clarify this impact.

**Questions:**

1. Regarding weakness 2, since PhysDreamer employs only an elastic component, does this imply that it performs significantly worse on hyper-elastic materials such as fluids?
2.  Beyond video-quality metrics, I wonder if it's possible to assess the performance of physical property estimation on synthetic scenes?
3. The authors compare with DreamGaussian4D in the qualitative results, yet it does not appear in the quantitative results. Could the authors clarify this choice?

---

### Official Review · Reviewer_UJoR · 2024-11-04

**Soundness:** 3
**Presentation:** 3
**Contribution:** 3
**Rating:** 5
**Confidence:** 4

**Summary:**

Physics3D proposed novel physics dynamics for 3D Gaussians based on an elastoplastic and a viscoelastic material model, which are simulated by MPM.
Using the SDS method, Physics3D extracts physical priors from a video diffusion model to identify the fundamental physical properties that govern object behavior.

**Strengths:**

1. The writing of the paper is comprehensive.
2. The material modeling in Physics3D is more general than other existing methods.
3. Physics3D integrates the physical properties of viscoelastic materials into the MPM for 3D Gaussian with physics dynamics, which is a great technical contribution.
4. The learnable internal filling strategy is effective.

**Weaknesses:**

1. Since Physics3D is more focused on general material modeling, the author should first provide the quantitative comparisons and videos of different types of simple dynamics demonstrated in Supp. B.2. The reviewer believes Physics3D can get better results. However, these simple corner cases with comprehensive comparisons can make Physics3D more convincing.
2. Although the quantitative results show that Physics3D can achieve the best visual quality, the reviewer thinks the author should visualize the material property of the fitting results to demonstrate the plausible material distribution learned by Physics3D.

**Questions:**

1. Could the authors break down the quantitative results into each sample? Since different real-world test examples and synthetic samples can represent various types of material, arranging the quantitative results into a table at the instance level can demonstrate the superiority of Physics3D.
2. Could the authors add Frechet Video and Inception Distance metrics for quantitative evaluation following PhysDreamer[a]? The metrics used in the paper are low-level pixel-wise error evaluations. However, all the existing test samples do not have too many dynamics, and most of the pixels in the background are static, which leads to marginal quantitative improvement in PSNR and SSIM. Maybe the high-level semantic evaluation results can make the performance of Physics3D more convincing and distinguishable.
3. The reviewer is a little concerned about whether the material modeling will degrade the capacity of the visual modeling of 3D Gaussian. According to the `carnation` result in Fig.5, only Physics3D fails to model the white texture of the left curtain. So, the reviewer suggests that the authors can compare synthetic datasets, e.g., fitting the same object model with different texture mapping (like checkboard) by different methods.
4. Could the authors add the quantitative ablation of the learnable internal filling strategy in PSNR/SSIM/FID/FVD? The authors should provide sufficient results to ensure the effectiveness of their contribution.
5. How sensitive is Physics3D to the simulation sub-step compared with other martial modeling? If the elastoplastic + viscoelastic can achieve more robust dynamics, the improvement in training efficiency can also contribute to Physics3D. Perhaps the authors could test this on substep durations of 1e-3 to 1e-5 seconds.
6. Could the authors add the material visualization following PhysDreamer[a] by using heat maps? Although the quantitative results show that Physics3D can achieve the best visual quality, the reviewer thinks the author should visualize the material property of the fitting results to demonstrate the plausible material distribution learned by Physics3D.

Reference:
[a] Zhang, Tianyuan, et al. "Physdreamer: Physics-based interaction with 3d objects via video generation." European Conference on Computer Vision. Springer, Cham, 2025.

**Details Of Ethics Concerns:**

The authors claimed they certify that no URL (e.g., GitHub page) could be used to find the authors' identities.
However, in Sec.5.2 L466, the URL provided by the authors is not anonymous.
The webpage contains all authors' identities.

---

### Official Review · Reviewer_C91R · 2024-11-04

**Soundness:** 3
**Presentation:** 3
**Contribution:** 1
**Rating:** 1
**Confidence:** 5

**Summary:**

The paper introduces a framework for inferring the physical properties of a 3DGS reconstructed scene by leveraging video model priors. This approach builds upon PhysDreamer by Zhang et al. (2024) by incorporating more complex MPM simulations and utilizing video-based SDS. Given a boundary condition, the 3DGS is deformed following PhysGaussian Xie et al. (2024) using MPM simulation. The rendered video by 3DGS renderer is then fed into Stable Video Diffusion to evaluate SDS. The SDS loss guides the optimization of physical parameters such that the generated video matches the reality learned by the pretrained SVD model.

**Strengths:**

The presentation of the paper is clear.

**Weaknesses:**

The contributions of this work appear limited, with only two main differences from PhysDreamer by Zhang et al. (2024): incorporating viscoelasticity into the MPM simulation and using SDS instead of reference videos to optimize physical parameters. However, viscoelasticity modeling itself is not a novel contribution (see my question below). It remains unclear whether the observed improvements stem primarily from the use of a more sophisticated simulator or from the SDS optimization. Since the primary contribution here is the use of SDS optimization, it is essential to provide evidence that the improvements are indeed due to SDS optimization rather than the simulator's enhanced capabilities.

**Questions:**

- Is there any novelty in the viscoelasticity modeling? To my understanding, it appears identical to the approach in [1]. The 1D rheological model depicted in Figure 2 and the 3D finite strain multiplicative decomposition of the deformation gradient in Figure 3 align precisely with Figure 4 in [1]. However, this framework for modeling viscoelasticity is claimed as a contribution here, which does not seem appropriate.

- The elastoplastic modeling (Line 281 - 300) is exactly copied from [2]: the choice of base elastic model (fixed-corotated elasticity), the notation of $\theta_c, \theta_s$ are identical. But there is no reference to [2] included there.

- The viscoelastic modeling (Line 306-323) is exactly copied from [1]: the choice of base elastic model (stVK elasticity), the expression of trial strain $\epsilon_N$ are identical. But there is no reference to [1] included there.

[1] Fang, Yu, et al. "Silly rubber: an implicit material point method for simulating non-equilibrated viscoelastic and elastoplastic solids." ACM Transactions on Graphics (TOG) 38.4 (2019): 1-13.
[2] Stomakhin, Alexey, et al. "A material point method for snow simulation." ACM Transactions on Graphics (TOG) 32.4 (2013): 1-10.

---

### Note · Authors · 2024-11-15

I have read and agree with the venue's withdrawal policy on behalf of myself and my co-authors.